# The Detection of Stem-Like Circulating Tumor Cells Could Increase the Clinical Applicability of Liquid Biopsy in Ovarian Cancer

**DOI:** 10.3390/life11080815

**Published:** 2021-08-11

**Authors:** Snezhanna O. Gening, Tatyana V. Abakumova, Dina U. Gafurbaeva, Albert A. Rizvanov, Inna I. Antoneeva, Regina R. Miftakhova, Andrey B. Peskov, Tatyana P. Gening

**Affiliations:** 1Department of Human Physiology and Pathophysiology, Ulyanovsk State University, 42 Leo Tolstoy St., 432017 Ulyanovsk, Russia; taty-abakumova@yandex.ru (T.V.A.); abp_sim@mail.ru (A.B.P.); Naum-53@yandex.ru (T.P.G.); 2OpenLab ‘Gene and Cell Technologies’, Kazan (Volga Region) Federal University, 18 Kremlyovskaya St., 420008 Kazan, Russia; dinagafurbaeva@gmail.com (D.U.G.); rizvanov@gmail.com (A.A.R.); regina.miftakhova@gmail.com (R.R.M.); 3Ulyanovsk Regional Clinical Oncology Center, 90 12th September St., 432048 Ulyanovsk, Russia; aii72@mail.ru

**Keywords:** ovarian cancer, stem cells, circulating tumor cells, CTCs, CD44, CD133, ALDH, EpCAM, vimentin, cytokeratin

## Abstract

Stem properties allow circulating tumor cells (CTCs) to survive in the bloodstream and initiate cancer progression. We aimed to assess the numbers of stem-like CTCs in patients with ovarian cancer (OC) before treatment and during first-line chemotherapy (CT). Flow cytometry was performed (Cytoflex S (Beckman Coulter, CA, USA)) using antibodies against CD45; epithelial markers EpCAM and cytokeratin (CK) 8,18; mesenchymal vimentin (vim); and stem-like CD44, CD133 and ALDH. This study included 38 stage I–IV OC patients (median age 66 (Q1–Q3 53–70)). The CK+vim- counts were higher (*p* = 0.012) and the CD133+ALDHhigh counts were lower (*p* = 0.010) before treatment in the neoadjuvant CT group than in the adjuvant group. The patients with ascites had more CK+vim- cells before treatment (*p* = 0.009) and less EpCAM-vim+ cells during treatment (*p* = 0.018) than the patients without ascites. All the CTC counts did not differ significantly in paired samples. Correlations were found between the CK-vim+ and CD133+ALDHhigh (r = 0.505, *p* = 0.027) and EpCAM-vim+ and ALDHhigh (r = 0.597, *p* = 0.004) cells before but not during treatment. Multivariate Cox regression analysis showed that progression-free survival was longer with the presence of surgical treatment (HR 0.06 95% CI 0.01–0.48, *p* = 0.009) and fewer CD133+ALDHveryhigh cells (HR 1.06 95% CI 1.02–1.12, *p* = 0.010). Thus, CD133+ALDH+ CTCs have the greatest prognostic potential in OC among the phenotypes studied.

## 1. Introduction

Ovarian cancer (OC) is one of the most deadly malignancies, with a five-year overall survival rate ranging from 36 to 46.2% [1]. Treatment of OC remains challenging, and many experts attribute this to the tumor heterogeneity at the cellular and molecular levels [2]. In daily practice, OC prognosis assessment is still based on the clinical characteristics rather than the biological features of tumor progression [3,4]. 

It is generally accepted that ovarian cancer spreads predominantly by implantation, involving the peritoneum and relapsing within the abdominal cavity. However, circulating tumor cells (CTCs) were detected in the blood of OC patients already in the 19th century [5]. In the last decade, a growing body of evidence has demonstrated that CTCs are one of the local tumor progression sources [6,7]. Circulating tumor cells have also attracted much interest as an object of tumor heterogeneity research, showing both similarities and differences compared with the parenchyma cells of primary tumor tissue [8]. The study of CTCs has acquired much broader implications than it had done initially.

The biological as well as the biomarker role of CTCs in ovarian cancer is still debatable, in contrast to a number of other malignancies in which these cells have demonstrated prognostic utility and are actively studied [9,10]. This is possibly due to the coexistence of CTCs that are more or less aggressive, prone to prolonged dormancy or active proliferation, motile or incapable of moving [11]. In the context of the epithelial-mesenchymal transition, the phenotype of a malignant epithelial cell can be epithelial, mesenchymal or intermediate (EM). All these phenotypes differ from each other both in the expression of surface markers and in the peculiarities of the cell behavior [12,13,14]. Moreover, the vast majority of CTCs die in the bloodstream without implementing their malignant potential [15]. 

Cancer stem cells appear to be the most resistant to unsuitable environmental conditions. Such cells avoid anoikis, tend to “wait out” an unfavorable period of chemotherapy in the state of dormancy or autophagy and can independently reproduce the entire tumor structure while maintaining their own number in the case of a successful introduction into the target tissue [16,17,18]. Initially, researchers considered that stemness is associated with the mesenchymal phenotype, but nowadays, the evidence of the extremely high plasticity of cancer stem cells and the possible combination of stem properties with both epithelial and mesenchymal characteristics is compelling [19,20]. Identification of tumor stem cells is carried out indirectly in most studies, by evaluating the expression of characteristic markers [21], as other techniques are less accessible and more costly in terms of the potential translation into clinical practice. Cancer stem cells are present in OC tissues [22] and ascitic fluid [23], but we have found no data on the circulating stem cell counts in patients with different disease courses in the available literature.

The aim of our study was to assess the numbers of stem-like circulating tumor cells in patients with ovarian cancer before and during treatment in connection with clinical characteristics.

## 2. Materials and Methods

### 2.1. The Study Cohort

This study prospectively included patients (*n* = 38) with newly diagnosed FIGO stage I–IV OC, who were treated at the Ulyanovsk Regional Clinical Oncology Center in 2019–2020, according to the criteria. The inclusion criteria were histological or cytological verification of the diagnosis, absence of concomitant acute or chronic exacerbated diseases of any etiology and absence of other synchronous or metachronous developing malignant tumors. Voluntary informed consent was obtained from all the study participants in accordance with the principles of the Declaration of Helsinki (2013). The study was approved by the Ethical Committee of IMEPC of Ulyanovsk State University (protocol № 3 of 15 March 2018). 

The treatment strategy for each patient was discussed by a multidisciplinary team. In accordance with international clinical guidelines, all patients received chemotherapy (CT)—carboplatin AUC 6 + paclitaxel 175 mg/m^2^ IV at day 1 q3weeks for a total of 6 cycles. Depending on tumor resectability and operability at the time of diagnosis, patients received adjuvant CT after the standard cytoreductive surgery, or neoadjuvant CT with subsequent response evaluation and surgery after the 3rd cycle if an objective response was achieved. Patients also received maintenance therapy with bevacizumab 7.5 mg/kg q3weeks for a total of 12 cycles or until progression after first-line CT, if indicated and available. Patients who did not receive maintenance therapy underwent a close follow-up (Figure 1).

### 2.2. Enumeration of Circulating Tumor Cells

Blood samples were obtained twice: before anticancer treatment and after 3 cycles of adjuvant or neoadjuvant chemotherapy (in the case of neoadjuvant CT, the second sample was obtained before surgery). Flow cytometry (Cytoflex S (Beckman Coulter, Brea, CA, USA) with CytExpert Software) was used to estimate the numbers of CTCs with epithelial, mesenchymal, EM and stem-like phenotypes. Negative selection of all CTCs was based on the absence of expression of the leukocyte antigen CD45. We also used antibodies against the following antigens: epithelial adhesion molecule EpCAM (CD326); cytokeratin (CK) 8,18; mesenchymal cell marker vimentin; OC stem cell markers CD44, CD133 and ALDH in various combinations; isotypic control for all antibodies except ALDH, which had an internal control in the kit (Table 1).

Firstly, the number of non-leukocyte-like cells (CD45-) was estimated in each sample according to the gate set using the CD45 single staining control. Within the CD45- population, the numbers of cells were assessed on the graph by SSC-A, and the levels of fluorescence were assessed using the gate set with the single staining control. A cell was considered epithelial in the case of CD45 negativity and the expression of CK and EpCAM (CK+EpCAM+, EpCAM+vimentin-, CK+vimentin-); mesenchymal in the case of vimentin expression (vimentin+, EpCAM-vimentin+, CK-vimentin+); EM in the case of CK+vimentin+; and stem-like if it was positive for stem markers in different combinations (CD45-CD44+, CD45-CD133+, CD45-CD44+CD133+, CD45-ALDHhigh, CD45-ALDHveryhigh, CD45- CD44+ALDHhigh, CD45-CD44+ALDH+veryhigh, CD45-CD133+ALDHhigh, CD45-CD133+ALDHveryhigh, CD45-CD44+CD133+ALDHhigh, CD45-CD44+CD133+ALDHveryhigh). ALDH expression was subdivided into high and very high because of the presence of a separate bright-positive population: according to the fluorescence intensity, the CD45- population was divided into ALDHhigh (the cells located in the middle in the FITC channel, within the gate established using the control tube with DEAB) and ALDHveryhigh (the cells located to the right). These gates were used for the analysis (Figure 2).

### 2.3. Statistical Processing

Statistical analysis was performed using Statistica 13.5.0 software (TIBCO Software Inc., Carlsbad, CA, USA). Graphical representation of the data was obtained using GraphPad Prism. The normality of the distribution of quantitative data was checked using the Shapiro-Wilk criterion. The Mann-Whitney U test was used to compare independent subgroups. Relationships between nominative variables were assessed using Fisher’s exact criterion. The Wilcoxon W-criterion was used to compare the CTC counts before and during treatment. To assess correlations, the nonparametric Spearman coefficient interpreted by the Cheddock scale was used. Simple linear regression was also used to assess relationships between two quantitative variables. 

As the interval from the last administration of platinum-containing CT to relapse or progression is recognized as a criterion for OC platinum sensitivity according to GCIG4th [24], we divided the patients into 2 subgroups: platinum-sensitive (interval is more than 6 months) and non-platinum-sensitive (interval is less than 6 months); binomial regression was used to assess the prediction of sensitivity. We also analyzed predictors of progression-free survival using Cox multivariable analysis. Differences in all cases were considered statistically significant at *p* ≤ 0.05. 

## 3. Results

The median age of the patients at diagnosis was 66 (IQR, Q1–Q3 53–70) years. Most patients in our cohort (58%) had FIGO stage III disease and peritoneal effusion (82%) at the time of initial diagnosis. Half of the patients in the cohort (53%) were amenable to cytoreductive surgery during first-line therapy: 7 patients received it as a primary treatment followed by adjuvant CT, and 13 patients received it after the third cycle of neoadjuvant CT. Some of the patients (29%) received bevacizumab maintenance after the first-line chemotherapy completion.

The clinical characteristics of the patients are presented in Table 2.

When comparing the numbers of CTCs before and during treatment, no statistically significant differences were found among all the phenotypes studied (Table 3).

Cells coexpressing all three stem markers studied were detected before treatment only in six patients; they appeared in four more patients during treatment. As such cells are more likely to represent the stem cell subset than those expressing one or two markers, we present the detailed clinical characteristics of these participants separately in Table 4.

As shown in Table 4, 9 of the 10 patients who were positive for CD44+CD133+ALDH+ CTCs had ascites and unresectable disease at initial diagnosis, and among those who underwent histological examination, only one patient had non-serous cancer; this observation may be due both to the predominance of patients with unfavorable clinical characteristics in our cohort and to the association between the tumor extent within the abdominal cavity and stem cell shedding.

The number of epithelial CK+vim- cells was higher in the neoadjuvant CT-treated patients than in the adjuvant CT group (*p* = 0.012) (Figure 3a), whereas the number of CD133+ALDHhigh stem-like cells was lower in NACT than in ACT (*p* = 0.010) (Figure 3b). The number of epithelial CK+vim- cells before treatment was higher among patients with ascites at initial diagnosis than among those without ascites (*p* = 0.009) (Figure 3c). The number of mesenchymal EpCAM-vim+ cells during treatment was slightly lower (*p* = 0.018) in patients with ascites (Figure 3d).

Patients with a clinically platinum-sensitive first relapse had more EpCAM+CK+ epithelial cells during treatment (*p* = 0.051) but not before treatment. Among those with non-sensitive recurrence, more patients did not undergo surgical intervention than among those with sensitive recurrence (*p* = 0.002). Additionally, all patients who received bevacizumab maintenance therapy did not develop relapse earlier than 6 months after CT completion, in contrast to those who did not receive bevacizumab (*p* = 0.008). This observation is consistent with the results of bevacizumab treatment demonstrated in landmark studies [25].

In the pre-treatment samples, we observed positive Spearman correlations between the counts of EpCAM+CK+ and EpCAM-vim+ (r = 0.427, *p* = 0.021), EpCAM+CK+ and vim+ (r = 0.436, *p* = 0.018), EpCAM+vim- and EpCAM-vim+ (r = 0.384, *p* = 0.040), CD44+ALDHveryhigh and CK+vim- (r = 0.377, *p* = 0.048), CK-vim+ and CD133+ALDHhigh (r = 0.505, *p* = 0.027) and EpCAM-vim+ and ALDHhigh (r = 0.597, *p* = 0.004) (Figure 4).

No statistically significant correlations between the cell numbers of different phenotypes were observed during treatment, except for a negative correlation between EpCAM+CK+ and CD44+CD133+ (r = −0.584, *p* = 0.011). In our cohort, there was a correlation between the white blood cell count and CD44+ALDHveryhigh cells before treatment (r = 0.419, *p* = 0.037); counts of CD133+ cells were not associated with the number of white blood cells. This may indirectly reflect a lack of CD133 expression by leukocytes, making this marker more attractive for liquid biopsy compared with CD44, which is often present on both cancer stem cells and leukocytes [26]. We also observed a negative correlation between CK+vim+ cell counts and CA-125 levels before treatment (r = −0.417, *p* = 0.034). In our cohort, however, serum CA-125 levels were not significantly associated with any clinical characteristics.

Interestingly, in samples obtained before treatment, the number of CD44+CD133+ALDHhigh cells was in a linear regression relationship with the number of EpCAM+CK+ cells (R^2^ = 0.315, *p* = 0.002). This relationship did not persist during treatment.

In a binomial regression analysis for the likelihood of platinum-sensitive vs. non-sensitive recurrence, a higher number of epithelial EpCAM+CK+ CTCs before treatment were nearly significantly associated with a higher probability of platinum-sensitive recurrence (OR 1.60, 95% CI 0.996–2.57) combined with the presence of surgery (OR 40.84, 95% CI 2.78–599.1) and age (OR 1.22, 95% CI 1.03–1.43).

In the Cox multivariable analysis, completion of surgical treatment (HR 0.06 95% CI 0.01–0.48, *p* = 0.009) and fewer CD133+ALDHveryhigh cells (HR 1.06 95% CI 1.02–1.12, *p* = 0.010) were associated with prolonged progression-free survival (PFS) (R^2^ = 0.481, LR test = 17, overall model *p* < 0.0001). The median PFS reached 12 [95% CI 9.7–17.7] months for patients who underwent surgery and 4.9 [95% CI 3–8.6] months for those who did not; for the patients with a CD133+ALDHveryhigh count above the cut-off (six cells), the median PFS was 6.4 [95% CI 0.9–NA] months, while for those with a cell count below the cut-off, it was 9.7 [95% CI 8.6–NA] months. The median overall survival in our cohort has not yet been reached. The results of the univariable and multivariable analyses are shown in Table 5.

## 4. Discussion

We evaluated the counts of CTCs of different phenotypes, including stem-like phenotypes, in ovarian cancer patients before and during treatment. 

We found that the number of epithelial EpCAM+CK+ phenotype cells before treatment was higher in patients with unresectable vs. resectable tumors, and also in the presence of ascites vs. the absence of ascites. At the same time, patients who subsequently developed clinically platinum-insensitive recurrence had fewer epithelial CTCs during treatment compared with platinum-sensitive patients. Moreover, we observed a tendency for a greater number of epithelial CTCs before treatment to be a predictor of platinum-sensitive recurrence. In a broader assessment of progression-free survival (with Cox multivariate analysis), the numbers of epithelial CTCs did not demonstrate a predictive role.

The prognostic potential of epithelial (CD45-negative, cytokeratin-8, 18- and/or 19- and EpCAM-positive) CTCs in ovarian cancer was evaluated by several groups of researchers. Poveda et al. [27] determined the counts of epithelial CTCs in 216 relapsed OC patients before the treatment start. A CTC count of ≥2 was significantly associated with an increased risk of progression and death in the univariate analysis, but the association was borderline significant in the multivariate analysis when the established prognostic factors—platinum-free interval, CA-125 level, number and size of tumor foci, tumor grade and ECOG overall performance status—were considered. In another study including patients with advanced and relapsed OC, there was no association between the presence of epithelial CTCs and prognosis [28]. In primarily diagnosed ovarian cancer, the epithelial CTC counts were studied by Banys-Paluchowski et al. [29]. The authors performed the sampling three times: before treatment, and after the third and sixth cycles of first-line chemotherapy. The study included patients with primary and relapsed disease; in the whole cohort, the overall survival was lower among patients who were CTC-positive initially or at follow-up than among the CTC-negative women, but the association was not confirmed in the primary OC subgroup analysis. Possibly, the high number of epithelial CTCs in the blood prior to treatment reflects a more advanced disease but not the lack of platinum sensitivity. Epithelial malignant cells are not able to actively move towards a blood vessel and cleave the intercellular matrix, but with a large tumor volume, CTCs can enter the circulation passively due to the insufficiency of the blood vessel walls within a tumor, as well as the weakness of the intercellular contacts in cancerous tissue. The majority of OC tumors are initially platinum-sensitive and demonstrate a high response rate to platinum-containing CT. Thus, the inconsistency of the data on epithelial CTCs in ovarian cancer may be due to the fact that epithelial cells are only a part of the CTC pool, actively participating in the progression of treatment-naïve tumors.

In our cohort, moderate positive correlations were present between epithelial and mesenchymal cell counts (EpCAM+CK+ and EpCAM-Vim+, EpCAM+CK+ and vimentin+, EpCAM+vimentin- and EpCAM-vimentin+) before treatment, indicating their simultaneous presence in the bloodstream. Moreover, the numbers of cells of some stem-like phenotypes were positively (moderately to noticeably) correlated with the numbers of mesenchymal (CD44+ALDHveryhigh and CK+vimentin-, CK-vimentin+ and CD133+ALDHhigh, EpCAM-vimentin+ and ALDHhigh), but not epithelial, cells. We also found that the regression dependence of the number of stem-like CD44+CD133+ALDHhigh CTCs on the number of epithelial CTCs was present before treatment but disappeared during treatment. At the same time, there were no statistically significant differences in the counts before and during treatment for any of the CTC phenotypes. In a study by the OVCAD Consortium [30], the authors used a sophisticated approach to detect EpCAM+ ovarian cancer CTCs, which included parallel assessment with qPCR and immunofluorescent staining in blood samples obtained before and after primary treatment (6 months after the adjuvant CT completion). The CTC counts before and after treatment did not differ, but there was some concordance in the results between the methods in the pre-treatment samples, whereas after treatment, there were no concordant positive findings at all. Long-term survival in this study was associated with CTC positivity at follow-up, but not at baseline. These observations also suppose the simultaneous coexistence of different CTC subtypes and changes in the ratio of subtypes during therapy, which is especially relevant for stem cells with their high phenotypic plasticity.

Specific detection of mesenchymal CTCs is challenging. Po et al. [31] found that, by using mesenchymal markers or a combination of EM surface markers, a greater number of ovarian cancer CTCs can be detected than using epithelial markers alone. The authors isolated CTCs from the blood of 18 patients with ovarian cancer by an immunomagnetic method—using antibodies to EpCAM and mesenchymal N-cadherin. However, N-cadherin-positive cells were also present in the blood of the control group, resulting in a high probability of false positive results. Another study addressed the detection of CTCs using the mesenchymal marker vimentin [32] and also detected much higher counts of mesenchymal CTCs (defined as CD45-DAPI+vimentin+) than epithelial CTCs (EpCAM+). The authors emphasized that the source of vimentin in the blood can be leukocytes, since this protein is a component of their cytoskeleton. This leads to false positive test results in healthy donors [20]. In agreement with these results, in our study, the numbers of vimentin+ cells in the blood were also high; we detected significantly fewer (*p* < 0.001) cells when using the EM combination of CK+vimentin+ markers (median of 6 and 5 before and during treatment, respectively), but we observed no association with clinical characteristics for any of the vimentin-expressing cells. Probably, a sufficient accuracy of mesenchymal CTC isolation in ovarian cancer can be achieved by the combination of an immune-based enrichment method with other methods, or the combination of mesenchymal and stem markers.

A number of markers expressed on ovarian cancer stem cells have been described in the literature: CD133 (prominin), ALDH (aldehyde dehydrogenase), CD44 (hyaluronan), CD117 (c-kit), MyD88 (myeloid differentiation primary response protein), CD24 (mucin-like adhesion molecule) [33]. The CD44 activation in OC cells stimulates stem-associated signaling of Nanog-Stat-3 [34]. In addition, CD44+ cells secrete more cytokines such as IL-6, IL-8 and MCP-1 than CD44-. The secretion of inflammatory mediators may mediate interactions with the microenvironment and contribute to the formation of stem and metastatic niches. Coexpression of CD44/MyD88 in advanced OC is associated with a decrease in progression-free and overall survival [35]. As reported by Loreth et al. [36], CD44 is expressed on the majority of CTCs obtained from patients with breast cancer, non-small cell lung cancer and melanoma with brain metastases while being much less expressed in the metastatic tissue. The authors concluded that the appearance of CD44 on the CTC surface plays an important role in the trafficking of a viable cell through the circulatory bed. However, the presence of CD44+ CTCs was associated with a poorer overall survival only in patients with melanoma but not with other tumors in the cohort.

The stem marker CD133 is known to enhance the ability of ovarian cancer cells to attach to and infiltrate the peritoneal mesothelium. The expression of CD133 contributes to, although it may be not sufficient for, the acquisition of the stem phenotype by up-regulating a number of stem-related genes [37]. Some authors have observed an association between high CD133 expression in OC tissue and decreased relapse-free and overall survival [38], but others have not [39]. The expression of CD133 significantly overlaps with ALDH expression in fresh tumor cells obtained from ovarian cancer patients, and CD133+ALDH+ cells are more capable of sphere formation than their CD133- and ALDH- counterparts [40]. The association of high ALDH expression with the adverse clinical course of high-grade serous and clear-cell OC has been well demonstrated [41]. Chefetz I. et al., who studied the effects of an ALDH inhibitor on ovarian cancer cells, called ALDH “arguably the best-characterized CSC marker” [42]. In ovarian cancer, ALDH+ cells show low sensitivity to chemotherapy compared to ALDH- [43]. This may be associated with impaired regulation of the cell cycle and DNA repair in ALDH+ cells [44], or with the ALDH involvement in the processes of detoxification and elimination of oxidative stress [45], which can also provide the cell with an advantage when entering the bloodstream.

We did not observe an association between the numbers of cells expressing only CD133 or only ALDH and clinical characteristics, but a high number of stem-like CD133+ALDHhigh cells before treatment were associated with a reduced progression-free survival in a multivariable analysis. It has previously been shown that platelets invert the cytostatic effects of carboplatin and paclitaxel on CD133+ and ALDH+ OC cells by secreting a number of cytokines and growth factors, in non-adherent conditions [46]. Further research is needed to understand the mechanisms of the interaction between OC circulating stem cells and other blood components.

The limitations of our study include the small sample size including patients from only one clinical center; the use of a limited number of markers for stem-like cell detection; the unavailability of tumor tissue for pathological assessment in 34% of patients, due to which we did not specify the types of adenocarcinoma in these women; and the lack of confirmation of stem-like characteristics of detected CSCs in vitro—the last point, however, would have made the methodology significantly more complex and costly, whereas our goal was to find a direction for research having a potential for translation into clinical practice. Larger studies are needed to assess the prognostic potential of stem-like CTCs in ovarian cancer.

Thus, among the studied phenotypes, the high number of stem-like CTCs in ovarian cancer is the most unambiguously associated with unfavorable clinical characteristics. At the same time, the CTC counts before and during treatment do not differ significantly, but the statistical relationships between the different populations do change. Probably, stem-like CTCs have no clear affiliation with the epithelial or mesenchymal phenotype. The search for new combinations of stem-like markers to identify circulating stem cells in ovarian cancer may help to discover a new biologically justified prognostic marker to stratify patients at primary diagnosis.

## 5. Conclusions

The identification of CD133+ALDH+ stem-like circulating tumor cells is more promising for studies on tumor biology and prognosis in ovarian cancer than the identification of CTCs by epithelial and mesenchymal marker expression alone. The numbers of stem-like CTCs are positively correlated with the mesenchymal CTC numbers before treatment; however, the statistical relationships between the CTC counts change during the treatment course despite the maintenance of the numerical values. Further studies are warranted to assess the role of mesenchymal features in stem-like CTC dissemination and to clarify the contribution of CD133+ALDH+ CTCs to OC progression.

## Figures and Tables

**Figure 1 life-11-00815-f001:**
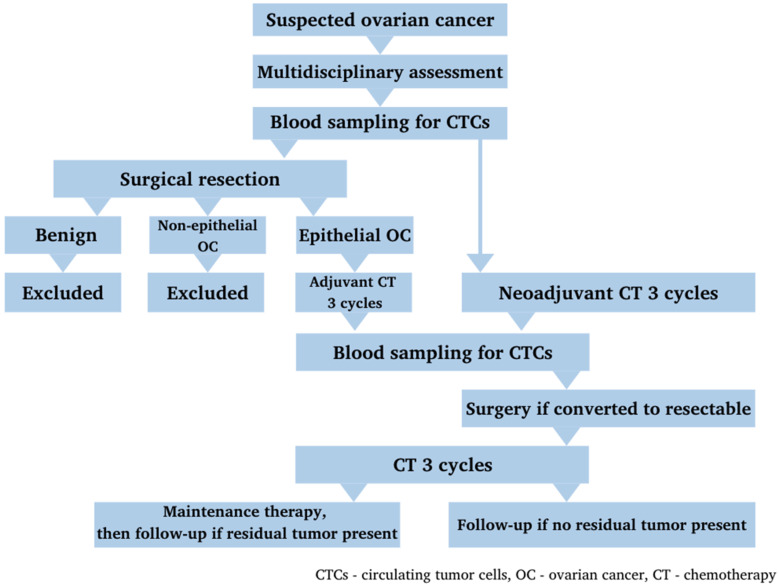
Patients’ selection, treatment and sampling within the study.

**Figure 2 life-11-00815-f002:**
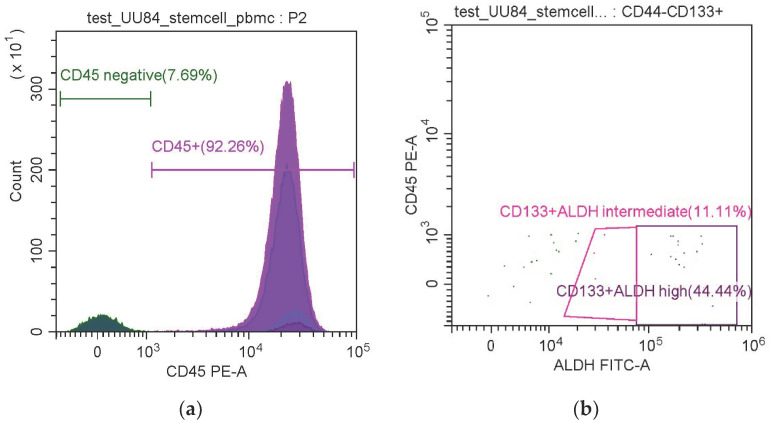
Identification of different populations of circulating tumor cells: (**a**) discrimination by CD45 binding intensity; (**b**) assessment of the ALDH expression intensity within the CD45-CD133+ population.

**Figure 3 life-11-00815-f003:**
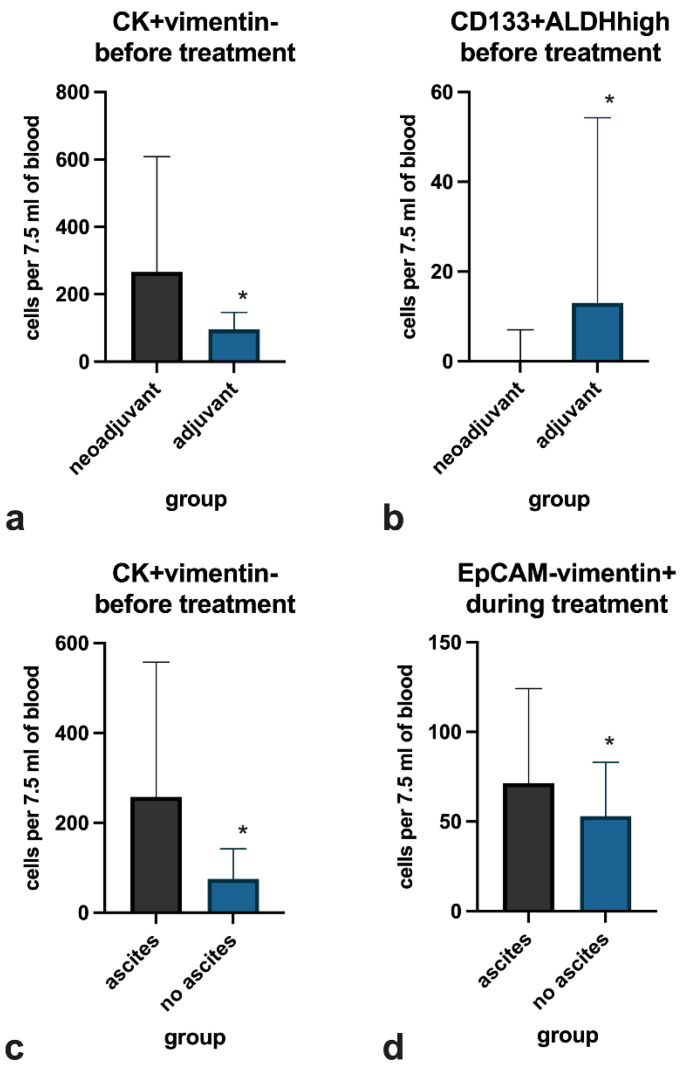
Statistically significant (*) differences in CTC counts between groups: (**a**,**b**) differences between groups of neoadjuvant vs. adjuvant therapy; (**c**,**d**) differences between groups with and without ascites. Data are presented as median and interquartile range.

**Figure 4 life-11-00815-f004:**
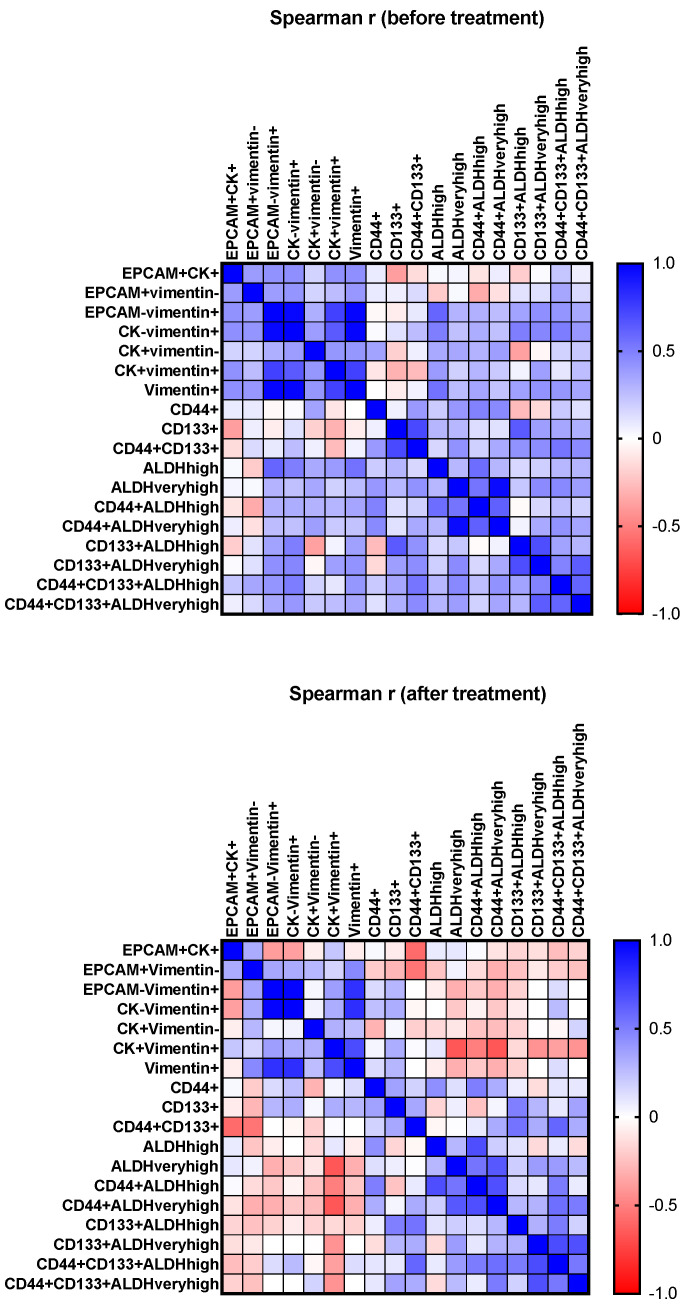
Correlation matrix for the CTC counts before and during treatment.

**Table 1 life-11-00815-t001:** Antibodies used in the analysis by flow cytometry.

No.	Manufacturer	Product	Usage
1	BioLegend, CA, USA	PE anti-human CD45 Antibody Clone 2D1	CD45
2	BioLegend	PE Mouse IgG1, κ Isotype Ctrl (FC) Antibody clone MOPC-21	Isotype
3	BioLegend	APC anti-human CD44 Antibody Clone BJ18	CD44
4	BioLegend	APC Mouse IgG1, κ Isotype Ctrl Antibody clone MOPC-21	Isotype
5	Miltenyi biotec, Bergisch Gladbach, Germany	CD133/1 VioBright 667 mouse IgG1 clone AC133	CD133
6	Miltenyi biotec	VioBright 515 mouse IgG1, Clone IS5-21F5	Isotype
7	Stemcell, Canada	ALDEFLUOR kit #01700	ALDH
8	Invitrogen, CA, USA	CD326 (EpCAM) Monoclonal Antibody PE-Cyanine7 clone 1B7	EpCAM
9	Invitrogen	PE-Cyanine7 mouse/IgG1, kappa clone P3.6.2.8.1	Isotype
10	Abcam, Cambridge, United Kingdom	Mouse Anti-Cytokeratin 8 + 18 + 19 antibody clone 2A4 (ab41825) IgG1	Primary
11	BioLegend	Brilliant Violet 421™ anti-mouse IgG1 Antibody clone RMG1-1	Secondary
12	BioLegend	Alexa Fluor 647 anti-Vimentin Antibody clone 091D3 Mouse IgG2a	Vimentin
13	BioLegend	Alexa Fluor 647 anti-mouse IgG2a clone RMG2a-62	Isotype

**Table 2 life-11-00815-t002:** Clinical characteristics of the patients included in the study.

Parameter	Number of Patients (%)
Age—median (IQR Q1–Q3), years—66 (53–70)	
Serum CA-125 before treatment—median (IQR), U/mL—863 (309–1595)	
FIGO stage	
I	2 (5%)
III	22 (58%)
IV	14 (37%)
Ascites at diagnosis	
yes	31 (82%)
no	7 (18%)
Histological subtype	
serous high-grade adenocarcinoma	16 (42%)
other subtypes	4 (11%)
not determined due to CRS 3	5 (13%)
no histological assessment (only cytological verification)	13 (34%)
Chemotherapy	
neoadjuvant	31 (82%)
adjuvant	7 (18%)
Cytoreductive surgery within first-line treatment	
performed	20 (53%)
not performed	18 (47%)
Maintenance therapy with bevacizumab after first-line CT	
performed	12 (32%)
not performed	26 (68%)

**Table 3 life-11-00815-t003:** Cell counts of the studied phenotypes in the blood of patients before and during treatment (data are presented as median (Q1–Q3) unless otherwise indicated; Before = before treatment, During = during treatment).

Phenotype	EpCAM+CK+ (Median (Min–Max))	EpCAM+Vimentin-	EpCAM-Vimentin+	CK-Vimentin+	CK+Vimentin-	CK+Vimentin+	Vimentin+
Before	0 (0–11)	25 (9–48)	59 (15–108)	50 (14–89)	218 (112–392)	6 (2–24)	59 (20–137)
During	1 (0–15)	28 (12–45)	57 (40–80)	48 (24–77)	217 (101–310)	5 (0–12)	79 (50–93)
**Phenotype**	**CD44+**		**CD133+**	**CD44+CD133+**	**ALDHhigh**	**ALDHveryhigh**	
Before	527 (121–1519)		101 (15–291)	1 (0–14)	90 (50–152)	12 (1–57)	
During	205 (120–339)		172 (57–391)	4 (0–23)	110 (37–154)	6 (0–9)	
**Phenotype**	**CD44+ALDHhigh**	**CD44+ALDH veryhigh**	**CD133+ALDHhigh**		**CD133+ALDHveryhigh (Median (Min–Max))**		
Before	15 (2–49)	7 (0–26)	0 (0–10)		0 (0–83)		
During	16 (6–49)	1 (0–5)	6 (0–26)		0 (0–300)		

**Table 4 life-11-00815-t004:** Detailed clinical characteristics of the patients positive for CD44+CD133+ALDHhigh and/or CD44+CD133+ALDHveryhigh cells in one or both blood samples.

No.	Age	Stage	Histology	CT Regimen	Ascites	Platinum Sensitivity	CD44+CD133+ ALDHhigh Count	CD44+CD133+ ALDHveryhigh Count
**Patients positive for the cells at baseline**								
1	61	IV	HGS	NACT	Yes	No	11	34
2	69	IV	HGS	NACT	Yes	Yes (M)	7	0
3	72	III	NS	NACT	Yes	No	2	2
4	70	III	NS	NACT	Yes	No	2	2
5	64	III	HGS	NACT	Yes	Yes	1	4
6	49	IV	HGS	NACT	Yes	No	5	0
**Patients positive for the cells during treatment only**								
7	37	IV	NS	NACT	Yes	Yes (M)	29	44
8	73	IV	HGS	NACT	Yes	Yes	1	1
9	52	III	HGS	NACT	Yes	Yes	4	7
10	57	IV	Clear cell	ACT	No	Yes	3	0

The table shows the highest number of cells from the two samples (before and during treatment); NS—non-specified, NACT—neoadjuvant chemotherapy, ACT—adjuvant chemotherapy, HGS—high-grade serous, M—received maintenance treatment.

**Table 5 life-11-00815-t005:** Results of uni- and multivariable Cox regression analyses for progression-free survival.

Category	Univariable Analysis, HR (95% CI)	Multivariable Analysis, HR (95% CI)
FIGO stage		
Stage III		
Stage IV	1.13 (0.61–2.10), *p* = 0.699	
Ascites		
No ascites		
Ascites present	2.07 (1.03–4.18), *p* = 0.042	1.65 (0.81–3.37), *p* = 0.169
Surgery in first line		
Surgery not performed		
Surgery performed	0.14 (0.03–0.67), *p* = 0.014	0.06 (0.01–0.48), *p* = 0.009
Chemotherapy		
Adjuvant regimen		
Neoadjuvant regimen	2.06 (1.00–4.25), *p* = 0.051	1.27 (0.47–2.86), *p* = 0.562
Age	1.01 (0.98–1.04), *p* = 0.458	
Initial serum CA-125	1.00 (1.00–1.00), *p* = 0.619	
Initial WBC count in blood	1.19 (1.01–1.41), *p* = 0.040	1.21 (0.91–1.62), *p* = 0.195
Maintenance after first line	0.28 (0.11–0.68), *p* = 0.006	0.27 (0.06–1.27), *p* = 0.096
CTC number before treatment		
EpCAM+CK+	1.01 (0.89–1.15), *p* = 0.856	
EpCAM+vimentin-	1.00 (1.00–1.01), *p* = 0.260	
EpCAM-vimentin+	1.00 (1.00–1.00), *p* = 0.456	
CK-vimentin+	1.00 (1.00–1.00), *p* = 0.484	
CK+vimentin-	1.00 (1.00–1.00), *p*=0.711	
CK+vimentin+	1.00 (0.98–1.01), *p* = 0.727	
CD44+	1.00 (1.00–1.00), *p* = 0.377	
CD133+	1.00 (1.00–1.00), *p* = 0.885	
ALDHhigh	1.00 (1.00–1.00), *p* = 0.375	
ALDHveryhigh	1.00 (1.00–1.01), *p* = 0.080	
CD44+CD133+	1.01 (0.99–1.03), *p* = 0.292	
CD44+ALDHhigh	1.00 (0.99–1.00), *p* = 0.160	
CD44+ALDHveryhigh	1.01 (1.00–1.02), *p* = 0.175	
CD133+ALDHhigh	1.01 (0.97–1.04), *p* = 0.764	
CD133+ALDHveryhigh	1.03 (1.01–1.06), *p* = 0.014	1.06 (1.02–1.12), *p* = 0.010

## Data Availability

The data presented in this study are available on request from the corresponding author.

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
