# Peer review of "The Detection of Stem-Like Circulating Tumor Cells Could Increase the Clinical Applicability of Liquid Biopsy in Ovarian Cancer"

_life, 2021, doi:10.3390/life11080815_

Round 1
Reviewer 1 Report
The study entitled “The detection of stem-like circulating tumor cells could increase the clinical applicability of liquid biopsy in ovarian cancer” is aimed to assess the numbers of stem-like CTCs in patients with ovarian cancer before treatment and during the first-line chemotherapy through flow cytometry analysis. The manuscript presents same limitations as evidenced by the authors and missing information. Making same minor revision can contribute in the understanding and tumor biology and development of prognosis in ovarian cancer based on stem-like circulating tumor cells.
In the first part of the results section information on the sample size under study are provided, I suggest moving this type of information to the materials and methods part and give more details about the patient collection
The numbering of the figures is wrong and this makes it difficult to understand the results; Moreover, in Figure 2 (Statistically significant differences in CTCs counts between groups:) for a better interpretation of the data presented I suggest to I suggest to reposition and enlarge the letters a) b) c) d) and show statistical significance on the columns of the histograms (asterisks or letters)
Table 4: only data from 10 patients are provided, but the sample analysed was largest; please clarified
Table 5: the column “Category” (especially the first few lines) it is not easy to understand;
Conclusions need to be increased by better detailing in the context of the limitations of the work and to have a more detailed insight into future perspectives
line 278: Authors reported that: “The study was not designed to evaluate correlations with clinical parameters and assess patient survival.” in my opinion it is a pity and a great limitation of the study, the authors must explain more the motivation for this choice or possibly compensate with analysis regarding their experimental evidence and individual clinical/not clinical (e.g.life stile etc..)parameters
Minor Comment:
Line 272: reference Po et al., is missing as number; please, check
Reviewer 2 Report
Gening et al. presented a detection of stem-like circulating tumor cells and its clinical applicability of liquid biopsy in ovarian cancer. Overall, the authors well described, however, there are some important concerns to be revised by the authors.
Major points;
Materials and Methods
1) The authors used FIGO classification to assess the ovarian cancer disease burden. However, the FIGO classification may lack accuracy in stratifying patient prognosis. Please reconsider to use other classifications such as WHO classification (Muallen et al. Cancers (Basel). 2021 Jul 2;13(13):3326. doi: 10.3390/cancers13133326.)
2) Although the authors stated that the Carboplatin +Paclictaxel therapy is the standard chemotherapy for ovarian cancer patients based on the international clinical guidelines, the dose of carboplatin may be AUC=6 (Kuzuya et al. Int J Clin Oncol. 2001 Dec;6(6):271-8. doi: 10.1007/s10147-001-8027-7.).
3) The patients’ selection flow is difficult to understand. Please describe the patients’ selection flow figure and show which subgroups were finally selected for analysis.
Result
1) The definition of ALDH high and veryhigh is not clear. Please describe the definition of the ALDH high and veryhigh.
2) According to the table 2, the patients included in the study contained large heterogeneity. The authors should separately perform analysis the patients with no histological assessment. It may be a confounder to the result (Akkiprik et al. Clin Breast Cancer. 2020 Aug;20(4):332-343.e3. doi: 10.1016/j.clbc.2020.02.006.)
3) In figure2, it seems that the flow cytometry was performed. However, the sequential gating should consider to be performed and the gating strategy should be described in the main text. (Kryczek et al. J Int J Cancer. 2012 Jan 1;130(1):29-39. doi: 10.1002/ijc.25967..)
4) The authors showed several correlation using Spearman’s coefficient. However, the correlation does not as strong as the author discussed later in the discussion section. The r=0.4-0.5 and r2=0.3-0.4 don’t represent that there is a strong correlation between the groups, even the statistical p-value was <0.05. If authors want to argue these correlations are strong, the scatter plots should be presented.
5) In table 5, the authors performed the statistical analysis using Cox multivariable analysis. However, the median progression free survival in each group was not presented in the main text.
Round 2
Reviewer 2 Report
This paper is a significant contribution, and I think the current revision can be accepted for publication.